# A New Contactless Cross-Correlation Velocity Measurement System for Gas–Liquid Two-Phase Flow

**DOI:** 10.3390/s23104886

**Published:** 2023-05-19

**Authors:** Bixia Sheng, Junchao Huang, Haifeng Ji, Zhiyao Huang

**Affiliations:** 1State Key Laboratory of Industrial Control Technology, College of Control Science and Engineering, Zhejiang University, Hangzhou 310027, China; shengbixia1993@163.com (B.S.); hfji@zju.edu.cn (H.J.);; 2Xiaoshan District Committee of the Communist Youth League, Hangzhou 311222, China; 3College of Optical, Mechanical and Electrical Engineering, Zhejiang A&F University, Hangzhou 311300, China

**Keywords:** cross-correlation velocity measurement, gas–liquid two-phase flow, small channel, contactless conductivity detection (CCD)

## Abstract

Based on the principle of Contactless Conductivity Detection (CCD), a new contactless cross-correlation velocity measurement system with a three-electrode construction is developed in this work and applied to the contactless velocity measurement of gas–liquid two-phase flow in small channels. To achieve a compact design and to reduce the influence of the slug/bubble deformation and the relative position change on the velocity measurement, an electrode of the upstream sensor is reused as an electrode of the downstream sensor. Meanwhile, a switching unit is introduced to ensure the independence and consistency of the upstream sensor and the downstream sensor. To further improve the synchronization of the upstream sensor and the downstream sensor, fast switching and time compensation are also introduced. Finally, with the obtained upstream and downstream conductance signals, the velocity measurement is achieved by the principle of cross-correlation velocity measurement. To test the measurement performance of the developed system, experiments are carried out on a prototype with a small channel of 2.5 mm. The experimental results show that the compact design (three-electrode construction) is successful, and its measurement performance is satisfactory. The velocity range for the bubble flow is 0.312–0.816 m/s, and the maximum relative error of the flow rate measurement is 4.54%. The velocity range for the slug flow is 0.161 m/s–1.250 m/s, and the maximum relative error of the flow rate measurement is 3.70%.

## 1. Introduction

The velocity is an essential parameter of the gas–liquid two-phase flow that describes the dynamic characteristic of the fluid [1,2,3]. Additionally, many works indicate that the velocity should also be considered for the measurement of other parameters of gas–liquid two-phase flow, such as the liquid film thickness, the void fraction, the flow rate, etc. [4,5,6,7,8,9,10,11] Small channels, known for their high surface-to-volume ratio and superior heat and mass transfer performance [4,5,6,12], have received increasing attention in various areas including Microelectromechanical Systems (MEMS), semiconductor radiators, micro/meso-scale chemical engineering, etc. [4,5,6,12] Therefore, for both academic research and industrial applications, online measurement of the velocity in small channels is of great importance [1,2,3,4,5,6,7,8,9,10,11].

The velocity measurement method based on the cross-correlation velocity measurement principle [13,14,15,16,17,18,19,20,21,22,23,24,25,26,27,28,29,30,31] is an effective method that has been applied and studied for many years [13,14,15,16,17,18,19,20,21,22,23,24,25,26,27,28,29,30,31]. The basic principle of cross-correlation is shown in Figure 1 [13,14,15].

As shown in Figure 1, an upstream sensor and a downstream sensor are installed on the measuring channel along the flow direction of the measured gas–liquid two-phase flow with a spacing L to obtain the upstream signal $x\left(t\right)$ and the downstream signal $y\left(t\right)$. Theoretically, a time delay $\tau^{*}$ only exists between $x\left(t\right)$ and $y\left(t\right)$, i.e., $y\left(t\right)=x\left(t+\tau^{*}\right)$. $\tau^{*}$ is also known as the transit time. The flow velocity v can be measured by
(1)$$v=k\frac{L}{\tau^{*}}$$
where k is the correction coefficient. According to the cross-correlation velocity measurement principle, to obtain $\tau^{*}$, the cross-correlation function $R_{xy}$ should be obtained. For continuous time signals [12,13,14],
(2)$$R_{xy}\left(\tau \right)=\frac{1}{T}\int_{0}^{T}x\left(t\right)y\left(t+\tau \right)dt$$
where T is the total duration of the signal. For discrete time signals [13,14,15],
(3)$$R_{xy}\left(i\right)=\frac{1}{N}\overset{N}{\underset{n=1}{\sum }}x\left(n\Delta t\right)y\left[\left(n+i\right)\Delta t\right]$$
where N is the total sequence length of the signal for the cross-correlation operation. The time corresponding to the peak position of $R_{xy}$ is the transit time $\tau^{*}$, and the peak position can be determined by a peak search.

Based on the principle of cross-correlation velocity measurement, upstream and downstream flow information can be obtained by different types of sensors [16,17,18,19,20,21,22,23,24,25,26,27,28,29,30,31], such as optical sensors [29,30,31], acoustic/ultrasonic sensors [17,18,27], and electrical sensors [16,19,20,21,22,23,24,25,26,28]. Velocity measurement with electrical sensors is applied by many researchers due to its advantages of good real-time performance, a simple sensor structure, and high safety [1,16,19,20,21,22,23,24,25,26,28]. Common electrical sensors include wire-mesh sensors [16], the conductivity probe [20], and ring conductivity electrodes [19,24]. However, in the existing research, velocity measurement is mainly implemented by contact electrical sensors, and when applied to small channels, it may suffer from the problems of electrochemical corrosion, electrode polarization, and flow interference [21,22,23].

The Contactless Conductivity Detection (CCD) sensor [32,33,34] is an effective electrical sensor for flow measurement in small channels due to its advantages of contactless measurement and excellent real-time performance [21,22,23,28]. Figure 2a,b shows the construction and the measurement principle of the CCD sensor [32,33,34].

As shown in Figure 2a, an insulated channel and two tubular metal electrodes installed on the outside of the channel make up the detection area of the CCD sensor. The two electrodes are the excitation electrode and the pick-up electrode. An AC source is applied to the excitation electrode, and a signal processing unit is used to obtain and process the signal output from the pick-up electrode [32,33,34]. The CCD sensor implements the contactless conductance measurement according to the measurement principle shown in Figure 2b. Under AC excitation, two coupling capacitances ($C_{1}$ and $C_{2}$) are formed between the two electrodes, the channel wall and the fluid. Further, a detection path is formed with $C_{1}$, $C_{2}$ and the equivalent conductance of the fluid $R_{x}$. By processing the output current *I_out_*, which contains the information about the $R_{x}$, with the signal processing unit, the measurement result of $R_{x}$ is achieved.

The CCD sensor shows great potential for use in the parameter measurement of gas–liquid two-phase flow in small channels [35,36,37,38,39,40,41]. Unfortunately, when trying to combine the CCD sensor with the cross-correlation velocity measurement principle and apply it to the velocity measurement of gas–liquid two-phase flow in small channels, difficulties exist [13,14,15]. Installing the upstream CCD sensor and downstream CCD sensor as shown in Figure 1 cannot implement effective velocity measurement due to the bad synchronicity, consistency, and independence of the upstream and downstream CCD sensors and the bad uniformity of the measured gas–liquid two-phase flow. More detailed information can be found in Section 2.

There have also been some reports on velocity measurement by CCD [21,22,23]. In current research, usually, four metal electrodes are installed to form two CCD sensors. For smaller pipe diameters (1–3 mm), four annular electrodes distributed along the axial direction are usually used [21,22]. For larger pipe diameters (3–6 mm), two pairs of radial electrodes are usually used [23]. The research results show that the axial four-electrode coverage area is too large [21,22], and the radial four-electrode is difficult to apply to smaller pipe diameters (less than 3 mm). At the same time, there is no existing solution to the problem of mutual interference between upstream and downstream sensors [21,22,23]. In addition, recent studies have indicated that, in small channels, the measured conductance signal of CCD sensors is affected by the fluid beyond the detection area, and the affected area is about 100 times the inner diameter of the channel [42]. This poses great challenges to the independence of upstream and downstream sensors.

In this work, to achieve the contactless velocity measurement of gas–liquid two-phase flow in small channels, a new contactless cross-correlation velocity measurement system with a three-electrode construction is developed. Firstly, the reason why CCD cannot be applied to cross-correlation velocity measurement is analyzed. According to the analysis, a system with a three-electrode construction that consists of an upstream/downstream switching unit, an upstream signal processing unit, and a downstream signal processing unit is developed. With the system, the synchronicity, consistency, and independence of the upstream CCD sensor and the downstream CCD sensor and the uniformity of the gas–liquid two-phase flow are guaranteed, and the velocity measurement of the gas–liquid two-phase flow is implemented. Two typical experiments are carried out to verify the effectiveness of the different parts of the developed system. Velocity measurement experiments of the actual slug flow and bubble flow are carried out to test the measurement performance of the developed system. Additionally, the uncertainty of the developed system is evaluated.

## 2. The Difficulties of Using CCD for Cross-Correlation Velocity Measurement

The difficulties of applying the CCD sensor for gas–liquid two-phase flow velocity measurement originate from the following two necessary conditions associated with the cross-correlation velocity measurement principle [13,14,15],

Condition 1 The measured fluid should be a solidified fluid, i.e., the relative positions between the gas slugs/bubbles should be stationary, and the shape of each slug/bubble should be fixed.Condition 2 The upstream sensor and the downstream sensor should be synchronous, consistent, and independent of each other.

The existing velocity measurement methods based on other types of sensors (such as optical sensors, acoustic/ultrasonic sensors, and contact electrical sensors) are mainly developed for normal-scale channels. The above methods can easily achieve a compact measurement area that satisfies Condition 1, and due to their measurement mechanism (strong directivity), synchronous, consistent, and independent measurement can be easily achieved, i.e., Condition 2 can also be satisfied. With the channel scale decreasing to small channels, the adaptability of the above measurement methods also decreases, while CCD sensors are suitable for measurement in small channels. However, when CCD is used for velocity measurement, it will face challenges in meeting the above two conditions.

### 2.1. The Reason Why Condition 1 Is Not Satisfied

Condition 1 is not satisfied with CCD sensors because of the long moving distance of the slugs/bubbles during the measurement process. Figure 3 shows the moving distance of the slugs/bubbles $d_{m}$, where the distance is the area covered by the upstream sensor and the downstream sensor.

As shown in Figure 3, $d_{m}$ is the sum of the electrode lengths and the gap length between the electrodes; if the electrode length is set to 20 mm and the gap between the electrodes is 10 mm, $d_{m}$ is 110 mm. However, with an increase in $d_{m}$, the possibility and degree of slug/bubble deformation and the relative position change will increase, which leads to Condition 1 not being satisfied. Figure 4 shows a typical example of deformation and relative position change.

From Figure 4a, it can be found that the bubble shape changes when it moves. Using the bubble marked by the red circle as an example, when it moves 43.0 mm, the deformation of the bubble is slight, and the gas–liquid two-phase flow can be regarded as solidified fluid. When it moves 85.9 mm, the deformation of the bubble can be observed by the naked eye. When it moves 130.5 mm, obvious deformation can be observed. In Figure 4b, the relative positions of the bubbles also change when they move. Using the circled three bubbles as an example, Table 1 lists the relative positions of the three bubbles.

From Table 1, it can be found that the relative positions of the three bubbles are continuously changing, and as the bubbles move, changes in relative position are accumulated. When the bubbles move 99.6 mm, the changes in the relative positions of the three bubbles are obvious.

### 2.2. The Reason Why Condition 2 Is Not Satisfied

Condition 2 is not satisfied due to the poor directional characteristics of CCD sensors. To ensure synchronicity, the upstream and downstream CCD sensors should work at the same time, but if they work at the same time, the upstream pick-up electrode receives the electrical signals from the upstream and downstream excitation electrodes as well as the downstream detection electrode. In this state, the consistency and independence of the upstream and downstream sensors is not satisfied. To better illustrate this problem, a finite element (FEM) simulation was carried out with COMSOL Multiphysics 5.5 software.

The finite element (FEM) simulation is an effective numerical simulation method that has been applied widely in the research field of multiphase flow. The construction of the developed FEM model and its computational grid are shown in Figure 5a,b. The model consists of two CCD sensors (an upstream CCD sensor and a downstream CCD sensor).

To simplify the solving process by reducing the model complexity, this paper employs a 3D finite element quasi-static field model to simulate and model the CCD sensors. The electrical characteristics of the CCD sensors within the field domain can be described by a set of equations based on Maxwell’s equations.
(4)$$\left\{\begin{array}{c} \nabla \cdot \left(\left(\sigma \left(x,y,z\right)+j2\pi f\varepsilon \left(x,y,z\right)\right)\nabla \phi \left(x,y,z\right)\right)=0\;\;\;\left(x,y,z\right)\subseteq \Omega \\ \phi_{a}\left(x,y,z\right)=V_{0}\;\;\;\;\;\;\;\;\;\;\;\;\;\;\;\;\;\;\;\;\;\;\;\;\;\;\;\;\;\;\;\;\;\;\;\;\;\;\;\;\;\;\;\;\;\;\;\;\;\;\;\;\;\;\;\;\;\;\left(x,y,z\right)\subseteq \Gamma_{a}\\ \phi_{b}\left(x,y,z\right)=0\;\;\;\;\;\;\;\;\;\;\;\;\;\;\;\;\;\;\;\;\;\;\;\;\;\;\;\;\;\;\;\;\;\;\;\;\;\;\;\;\;\;\;\;\;\;\;\;\;\;\;\;\;\;\;\;\;\;\;\;\left(x,y,z\right)\subseteq \Gamma_{b}\\ \frac{\partial \phi \left(x,y,z\right)}{\partial \vec{n}}=0\;\;\;\;\;\;\;\;\;\;\;\;\;\;\;\;\;\;\;\;\;\;\;\;\;\;\;\;\;\;\;\;\;\;\;\;\;\;\;\left(x,y,z\right)\subseteq \Gamma_{c},\left(c\neq a,b\right) \end{array}\right.$$
where $\sigma \left(x,y,z\right)$ and $\varepsilon \left(x,y,z\right)$ are the distribution functions of the electrical conductivity and relative permittivity of the medium within the field domain, respectively; f is the frequency of the AC excitation signal. $\phi \left(x,y,z\right)$ is the spatial distribution of the electric potential within the field domain, and $\phi_{a}\left(x,y,z\right)$ and $\phi_{b}\left(x,y,z\right)$ are the spatial distributions of electric potentials on the exciting electrodes and pick-up electrodes, respectively. $\Gamma_{a}$, $\Gamma_{b}$, and $\Gamma_{c}$ are the spatial locations of the exciting electrodes, pick-up electrodes, and floating electrode, respectively (the FEM model in Figure 5 does not contain floating electrodes, so $\Gamma_{c}$ is an empty set, but the FEM model employed in the next section contains floating electrodes, so the definition of floating electrodes is also provided). $V_{0}\;$ is the amplitude of the applied excitation signal, $\vec{n}$ is the unit vector in the direction of the external normal, and ∇ is the Hamiltonian differential operator.

Table 2 lists the properties of the model components.

The inner diameter of the small channel in the FEM model is 3.0 mm and the pipe wall is 0.5 mm in size. The material in the pipe wall is quartz glass (conductivity = 0 S/m and relative permittivity = 4.2). The electrodes are attached to the outer wall of the pipe, and the material is copper (conductivity =$\;5\times 10^{7}$ S/m). The fluid used is tap water (conductivity = 0.01 S/m and relative permittivity = 78). To simulate two CCD sensors working together, two of the electrodes (the two excitation electrodes from the two CCD sensors) were supplied with the excitation voltage and the other two were grounded. The amplitude and frequency of the excitation voltage were set to 2 V and 200 KHz. The length of the four electrodes was 20 mm, the gap between electrodes was 10 mm, and the length of the small channel was 200 mm.

The simulation result is shown in Figure 6. In Figure 6, the current density is represented by the red arrow.

As shown in Figure 6, when the excitation electrodes of the upstream and downstream CCD sensors were supplied with the excitation voltage, the pick-up electrode from the upstream CCD sensor obtained the electrical signal from the excitation electrodes of both the upstream and downstream CCD sensors. However, the pick-up electrode of the downstream CCD sensor obtained the electrical signal only from the excitation electrodes of the downstream CCD sensor. Thus, the consistency and independence of the upstream and downstream CCD sensor were not satisfactory.

### 2.3. The Challenges of Designing an Effective Velocity Measurement System Based on CCD

As mentioned in Section 2.1 and Section 2.2, to satisfy Condition 1, the moving distance of the bubbles/slugs should be short enough, i.e., the measurement area (the area covered by upstream and downstream sensors) of the velocity measurement system should be compact enough. The challenge is that the velocity measurement system requires two CCD sensors, so simply reducing the distance between the upstream and downstream sensors does not necessarily make the system more compact.

To satisfy Condition 2, the upstream and downstream CCD sensors should work synchronously, have consistent measurement characteristics, and their measurements should be independent. The challenge is that it is difficult to satisfy all three conditions of synchronicity, consistency, and independence at the same time for both the upstream and downstream sensors.

In summary, an effective CCD-based velocity measurement system should be compact enough and be able to satisfy the synchronicity, consistency, and independence requirements of the upstream and downstream sensors as much as possible.

## 3. The New Three-Electrode Velocity Measurement System

### 3.1. Construction of the Three-Electrode Velocity Measurement System

According to the above analysis, to apply the CCD technique to the cross-correlation velocity measurement of gas–liquid two-phase flow in small channels, a new velocity measurement system should be developed, and the new system should fit in with the following two technical specifications:(1)Compact construction,(2)The upstream CCD sensor and downstream CCD sensor can work independent,(3)Except for the positions of the measurement area, the other measurement characteristics of the upstream CCD sensor and downstream CCD sensor should be consistent.

In this work, a three-electrode velocity measurement system with a three-electrode construction was developed. The reason for designing a three-electrode construction was as follows: At least two identical sensors are required based on the principle of cross-correlation velocity measurement [13,14,15], and each CCD sensor must have at least two electrodes based on the CCD measurement principle [32,33,34]. Therefore, by reusing electrodes, the implementation of two CCD sensors using three electrodes is currently the solution that requires the fewest electrodes and the least space. Figure 7 illustrates the construction of the new system.

As shown in Figure 7, the velocity measurement system is composed of electrodes A, B, and C. In practical measurements, electrodes A and B constitute the upstream CCD sensor, while electrodes B and C constitute the downstream sensor. By reusing an electrode (electrode B), two CCD sensors are formed with only three electrodes, which greatly reduces the size of the system. The three-electrode construction of the new system is more compact than that of the system illustrated in Figure 3. If the electrode length is set to 20 mm, and the gap between electrodes is 10 mm, the moving distance of the slugs/bubbles of the three-electrode velocity measurement system is 80 mm, 30% less than that of the system illustrated in Figure 3.

To implement independent and consistent measurements of the upstream CCD sensor and downstream CCD sensor, a switching unit was specially designed. With the switching circuit, when electrodes A and B work as the upstream sensor, electrode C is floating, and when electrodes B and C work as the downstream sensor, electrode A is floating. Thus, consistent and independent measurement can be achieved. Meanwhile, to ensure synchronicity, the switching of the upstream sensor and the downstream sensor is fast enough. The upstream conductance measurement unit and the downstream conductance measurement unit convert the output current obtained by the upstream sensor and downstream sensor into conductivity measurement results and transmit them to the computer. The computer is introduced to process the conductivity measurement results and finally obtain the velocity measurement results.

As shown in Figure 7, the distance between the upstream sensor and downstream sensor is the distance between the centers of two excitation electrodes (or two pick-up electrodes), i.e., $l_{m}$ in Figure 7.

### 3.2. Switching Unit

Figure 8 shows the circuit connections in the switching unit.

As shown in Figure 8, the switching of the upstream sensor and downstream sensor is achieved by electronic switches K_1_–K_6_, which have the following two combination modes:

Upstream working mode: K_1_, K_2_, and K_6_ are connected, K_3_, K_4_, and K_5_ are disconnected, electrode A and electrode B work as the upstream sensor, electrode A is the excitation electrode connected to the AC source, and electrode B is the pick-up electrode, which is connected to the upstream conductance measurement unit to obtain the upstream conductance information. Electrode C is in a suspended state. The input of the downstream conductance measurement unit is connected to the ground, and the output is 0.

Downstream working mode: K_3_, K_4_, and K_5_ are connected, K_1_, K_2_ and K_6_, are disconnected, electrode B and electrode C work as the upstream sensor, electrode B is the excitation electrode connected to the AC source, and electrode C is the pick-up electrode, which is connected to the downstream conductance measurement unit to obtain the downstream conductance information. Electrode A is in a suspended state. The input of the upstream conductance measurement unit is connected to the ground, and the output is 0.

A sequence diagram of the switching unit is illustrated in Figure 9.

The electronic switch used in Figure 8 is an ADG433 electronic switch produced by Adno Semiconductor, Inc. It has the advantages of a small on–off resistance (<24 Ω), a short on–off time (all less than 165 ns), a small open-circuit leakage current (<25 pA), etc., and its performance fully meets the requirements. The ADG433 has two normally open and normally closed switches that can be independently controlled on each piece, so two ADG433 pieces were used in this work. The switching control signal of the upstream and downstream working modes is the same square wave signal, which can ensure a high degree of synchronization and improve the switching efficiency.

The FEM simulation was carried out with COMSOL Multiphysics 5.5 software to illustrate the current density distribution of the upstream working mode and the downstream working mode. Figure 10 show the current density simulation results of the upstream and downstream working modes, where (a) is the simulation model, (b) is the current density simulation result of the upstream working mode, and (c) is the current density simulation result of the downstream working mode. The electrical characteristics of the CCD sensors within the field domain here are the same as those in Section 2.2 and could also be described by Equation (4). For the upstream working mode, Electrode A is set as the exciting electrode, Electrode B is set as the pick-up electrode, and Electrode C is set as the floating electrode. For the downstream working mode, Electrode A is set as the floating electrode, Electrode B is set as the exciting electrode, and Electrode C is set as the pick-up electrode. The computational grid is set the same as that in Section 2.2.

As shown in Figure 10, when one sensor is working, the other electrode that does not participate is in the suspended state, and there is no electrical signal output or input on that electrode; thus, the measurement will not be interfered with. Thus, the independence and consistency of the upstream sensor and downstream sensor are ensured.

### 3.3. Conductance Measurement Unit

Figure 11 illustrates the conductance measurement unit.

The conductance measurement unit is composed of three modules, the simulated inductor module, the I/V circuit, and the analog phase sensitive demodulation (APSD) module, connected in series. The role of the simulated inductor module is to eliminate the adverse effects of the coupling capacitances $C_{1}$ and $C_{2}$ on the conductance measurement by the principle of series resonance [23,28,43,44], i.e., mutual elimination between the inductive reactance of the simulated inductor module and the capacitive reactance of the coupling capacitances $C_{1}$ and $C_{2}$. The impedance of the detection path and the simulated inductor module is
(5)$$Z=Z_{x}+j\left(2\pi fL_{eq}-\frac{C_{1}+C_{2}}{2\pi fC_{1}C_{2}}\right)+R_{eq}$$
where j is the imaginary unit, f is the excitation frequency, $Z_{x}$ is the equivalent impedance of the measured flow, and $Z_{x}=r_{e}+ji_{m}$, where $r_{e}$ is the real part, $i_{m}$ is the imaginary part, and $L_{eq}$ and $R_{eq}$ are the equivalent inductance and internal resistance of the simulated inductor module, respectively.

When the excitation frequency f is set to
(6)$$f=f_{0}=\frac{1}{2\pi }\sqrt{\frac{C_{1}+C_{2}}{L_{eq}C_{1}C_{2}}},$$
the following equation is satisfied,
(7)$$2\pi fL_{eq}=\frac{C_{1}+C_{2}}{2\pi fC_{1}C_{2}}$$
and Z is
(8)$$Z=Z_{x}+R_{eq}.$$

Thus, the capacitive reactance of the coupling capacitances $\frac{C_{1}+C_{2}}{2\pi fC_{1}C_{2}}$ is removed, and its adverse effects can be overcome. With the I/V circuit, $U_{out}\;$ is
(9)$$U_{out}=\frac{U_{in}}{Z_{x}+R_{eq}}R_{f}$$
where $R_{f}$ is the feedback resistance of the I/V circuit. It is necessary to indicate that, in this work, the simulated inductor module was introduced due to its adjustable characteristic [23,28,43,44]. According to Section 2.2, the consistency of the upstream sensor and the downstream sensor is necessary for velocity measurement. The adjustable characteristic of the simulated inductor ensures that the upstream and downstream sensors are more easily in a consistent working state.

The APSD module [45] was adopted to implement the measurement of the conductance $r_{e}$. By two multipliers, the $U_{out}$ is multiplied by the in-phase reference signal $U_{ro}$ ($U_{ro}=U_{in}$) and the orthogonal reference signal $U_{r90}$ ($U_{r90}$ is at the same frequency as $U_{in}$ and has a phase difference of 90°). Then, the high frequency components of the two multipliers’ outputs are filtrated with two low pass filters, the remaining DC parts $U_{0}$ and $U_{90}$ are obtained, and the real part $r_{e}$ and the imaginary part $i_{m}$ are [45]
(10)$$r_{e}=-\frac{P^{2}R_{f}}{2}\frac{U_{0}}{U_{0}{}^{2}+U_{90}{}^{2}}-R_{eq}$$
(11)$$i_{m}=\frac{P^{2}R_{f}}{2}\frac{U_{90}}{U_{0}{}^{2}+U_{90}{}^{2}}$$
where P is the amplitude of $U_{in}$. The measured conductance is the real part $r_{e}$. More detailed information about the conductance measurement unit can be found in References [23,29,43,44].

## 4. Velocity Measurement Experiment

### 4.1. Experimental Setup

To verify the effectiveness of the developed system, practical velocity measurement experiments of the bubble flow and slug flow in a small channel were carried out. A prototype of the new contactless cross-correlation velocity measurement system was developed, and an experimental setup was established. Figure 12 shows the developed experimental setup and a photo of the prototype. Table 3 lists the parameters of the prototype.

As shown in Figure 12, the prototype consists of a small channel with three electrodes, a switching unit, and two conductance measurement units, a data acquisition device, and a computer. In this work, the original signals of the system were sampled with a data acquisition device (cDAQ-9172, NI). The sampling frequency of the data acquisition device was 10.0 kHz. Then, the sampled data were transmitted to the computer to achieve cross-correlation and obtain the velocity. Figure 13a,b shows photos of the experimental bubble flow and slug flow captured by a high-speed camera.

The reference velocity $v_{\mathrm{ref}}$ was obtained by the high-speed camera by
(12)$$v_{\mathrm{ref}}=\frac{l_{c}}{N_{c}t_{c}}$$
where $l_{c}$ is the moving distance of the bubble/slug in two photos, $N_{c}$ is the number of flames between the two photos, and $t_{c}$ is the time interval between shots. Figure 14 provides an example of obtaining $v_{\mathrm{ref}}$ (using bubble flow (Figure 13a) as an example).

In Figure 14, the frame rate of the high-speed camera is 100 fps, $t_{c}=10ms$, $N_{c}=5$, and the moving distance of the bubble $l_{c}=34.5\;\mathrm{mm}$, according to (12), $v_{\mathrm{ref}}=0.70\;m$/s. The reference velocity of the slug flow in Figure 13b is 0.26 m/s.

### 4.2. Signal Processing

In practical measurements, the obtained original signals should be processed to obtain the velocity measurement result. Signal processing includes four steps: (1) signal extraction, (2) standardization, (3) sampling delay compensation, and (4) cross-correlation velocity measurement.

(1)
**Signal extraction**


The output signal of the system $u_{out}\left(t\right)$ is mixed with the switching process between the upstream and downstream signals, so it is necessary to extract the original signal. Figure 15 shows the output signal $u_{out}\left(t\right)$.

The switching between the upstream sensor and downstream sensor is controlled by a square wave signal $u\left(t\right)$. When $u\left(t\right)$ has a high voltage, the upstream sensor works, and when $u\left(t\right)$ has a low voltage, the downstream sensor works. The original signal of the upstream $u_{rout1}\left(t\right)$ and the original signal of the downstream $u_{rout2}\left(t\right)$ can be expressed as
(13)$$\begin{array}{cc} u_{rout1}\left(t\right) & =u\left(t\right)u_{upr}\left(t\right)\\ & =\left\{\right.\begin{array}{ll} u_{upr}\left(t\right) & NT<t\leq \left(N+D\right)T\\ 0 & \left(N+D\right)T<t\leq \left(N+1\right)T \end{array}\left(N=0,1,2\ldots \right) \end{array}$$
(14)$$\begin{array}{cc} u_{rout2}\left(t\right) & =u\left(t\right)u_{downr}\left(t\right)\\ & =\left\{\right.\begin{array}{ll} u_{downr}\left(t\right) & \left(N+D\right)T<t\leq \left(N+1\right)T\\ 0 & NT<t\leq \left(N+D\right)T \end{array}\left(N=0,1,2\ldots \right) \end{array}$$
where *T* is the period of the square wave and D is the duty ratio. All switching signals in this work are square waves with D = 50% and T=1 ms. $u_{upr}\left(t\right)$ is the original signal of the upstream signal, and $u_{downr}\left(t\right)$ is the original signal of the downstream signal. In a square wave period, the upstream sensor and downstream sensor work once each, and the working time for each sensor is 0.5 T.

At the same time, it can be seen from Figure 15 that it takes a certain amount of time to stabilize the signal during the switching process. Therefore, in order to carry out velocity measurements and extract effective upstream and downstream signals, only one signal is extracted from each switch when the original signal is preprocessed (that is, the sampling period of the upstream sensor and downstream sensor is T. In this work, the sampling period was 1 ms, and the sampling frequency was 1/T. In this work, the sampling frequency was 1 kHz) to obtain the original signal sequences of the upstream and downstream $U_{\mathrm{upr}}\left(n\right)$ and $U_{\mathrm{downr}}\left(n\right)$. Figure 16 shows the original signal sequences of the upstream and downstream $U_{\mathrm{upr}}\left(n\right)$ and $U_{\mathrm{downr}}\left(n\right)$.

(2)
**Standardization**


To ensure the effect of the subsequent cross-correlation, it is necessary to standardize the upstream and downstream signal sequences to the interval [−1, 1]. Standardization adopts maximum and minimum value standardization, and its process can be expressed as
(15)$$U^{*}{}_{upr}\left(n\right)=\left[\frac{2(U_{upr}\left(n\right)-U_{upr,min})}{U_{upr,max}-U_{upr,min}}-1\right]$$
(16)$$U^{*}{}_{downr}\left(n\right)=\left[\frac{2(U_{downr}\left(n\right)-U_{downr,min})}{U_{downr,max}-U_{downr,min}}-1\right]$$
where $U^{*}{}_{upr}\left(n\right)$ and $U^{*}{}_{downr}\left(n\right)$ are the standardized signal sequences of the upstream and downstream signals, $U_{\mathrm{upr},\min}$ and $U_{\mathrm{downr},\min}$ are the minimum values of the signal sequences of the upstream and downstream signals, and $U_{\mathrm{upr},\max}$ and $U_{\mathrm{downr},\max}$ are the maximum values of the signal sequences of the upstream and downstream signals. The standardized upstream and downstream signal sequences are shown in Figure 17.

(3)
**Cross-correlation velocity measurement**


From Figure 17, it can be seen that, for both the bubble flow and the slug flow, there are obvious correlation relationships between the upstream and downstream signal sequences, and there are time delays τ (i.e., transit time). As mentioned above, the correlation functions of the upstream and downstream signals can be obtained by (2) and (3), and the position corresponding to the maximum value of the correlation number is the transit time τ. Figure 18 shows the corresponding correlation number and the corresponding τ. According to τ, the velocity v to be measured can be obtained by the following formula:(17)$$v=k\frac{l_{m}}{\tau }$$

As shown in Figure 18, the transit times τ of the bubble flow and slug flow are 44 ms and 117 ms, respectively.

(4)
**Sampling delay compensation**


As Figure 6 shows, it is difficult for the upstream sensor and downstream sensor to ensure complete synchronicity, because electrical signals do not have the capability of directional transmission. If the upstream and downstream signals work together, the detection fields of the two groups of sensors will interfere with each other. Therefore, in each sampling, there is a time difference of 0.5 ms between the upstream and downstream signals, which is almost negligible compared with the common velocity segment (0–2 m/s, corresponding to a transit time of greater than 15 ms). However, to further ensure the measurement accuracy, this sampling delay calibration is compensated for in practical applications. That is, 0.5 ms is subtracted from the actual transit time for each acquisition. Each signal sequence is sampled, and the control signal starts at a high level (i.e., from the working upstream sensor), so the upstream signal obtained by each sampling is 0.5 ms ahead of the downstream signal, and compared with the obtained transit time τ, the corresponding actual transit time $\tau_{a}$ needs to compensate for this 0.5 ms, $\tau_{a}=\tau -0.5\;\mathrm{ms}$. Thus, the compensated transit times τ for bubble flow and slug flow are 43.5 ms and 116.5 ms, respectively.

(5)
**Theoretical velocity measurement limitations**


According to the signal processing method, the sampling interval is determined by the switching frequency of the switching unit. In this work, the sampling interval was 1 ms; thus, the time resolution of the transit time measurement was 1 ms. When the velocity of the bubble/slug is high, the transit time τ becomes very short, which can compromise the reliability of the velocity measurement result. Thus, there is an upper limit for the measured velocity $v_{\max}$, $v_{\max}=l_{m}/\tau_{\min}$, where $\tau_{\min}$ is the lower limit of the transmit time. For the prototype developed, $v_{\max}=30.0\;m$/s, which is much higher than the typical velocity of bubble flow and slug flow in small channels.

The lower limit is determined by the total duration used for calculating the cross-correlation function, i.e., T in Equation (2). If the velocity is too slow, the bubble/slug flow cannot entirely pass through the upstream and downstream sensors, and that will lead to inconsistency in the upstream and downstream CCD sensors. The total duration was 1.0 s, and the total detection area was 70.0 mm, so the lower limit of velocity measurement was $v_{\min}=0.07\;m$/s.

### 4.3. Experimental Result

In the above two typical experiments of the bubble flow and slug flow velocity measurement, the velocity of bubble flow was 0.705 m/s and the velocity obtained by the system of bubble flow was 0.690 m/s. The absolute measurement error was 0.015 m/s, and the relative measurement error was 2.13%. The velocity of the slug flow was 0.263 m/s, and the velocity obtained by the system of slug flow was 0.258 m/s. The absolute measurement error was 0.005 m/s, and the relative measurement error was 1.90%. The results of the two typical experiments show that the design of the switching unit and the conductance measurement unit is successful and can achieve effective sensor switching and conductance measurements, respectively. Additionally, the velocity measurement can be implemented with the principle of cross-correlation velocity measurement.

To verify the velocity measurement performance of the developed system, velocity measurement experiments of the bubble flow and slug flow were carried out with the prototype. Figure 4 shows the experimental results for the bubble/slug velocity measurement.

As shown in Figure 19, the velocity range for the bubble flow was 0.312–0.816 m/s, and the maximum relative error of the flow rate measurement was 4.54%. The velocity range for the slug flow was 0.161–1.250 m/s, and the maximum relative error of the flow rate measurement was 3.70%. From Figure 19, it can be found that for both flow patterns, when the velocity was higher, the measurement errors tended to be larger. The reason for this is that when the velocity is high, the transit time is short, and in this work, the resolution of the transit time measurement was 1 ms, so the error caused by the resolution increased. For example, if the transit times are 100.5 ms and 20.5 ms, due to the limitation of resolution, the measured transit times will be 100 ms and 20 ms, and the corresponding measured velocities will be 0.300 m/s (the real velocity is 0.299 m/s; the relative error is 0.33%) and 1.5 m/s (the real velocity is 1.463 m/s; the relative error is 2.60%), respectively. The influence of the limitation of resolution on the velocity measurement is more obvious at high velocities.

In addition, the uncertainty of the developed system was evaluated. Repeated experiments of velocity measurement were carried out 20 times for the same gas–liquid two-phase flow. The standard deviation σ of the repeated experiments was introduced as the uncertainty index. The determination of σ is carried out with
(18)$$\sigma =\sqrt{\frac{\sum_{i=1}^{n}{\left(v_{i}-\bar{v}\right)}^{2}}{n-1}}$$
where $v_{i}$ is the measured velocity of the *i*th repeated experiment, n is the number of repeated experiments, and $\bar{v}$ is the average velocity of the *n* groups of repeated experiments. The standard deviation of the 20 velocity measurement results was calculated and is presented in the form of error bars in Figure 19. The maximum standard deviation of the measured velocity for the bubble flow was 0.012 m/s, and the maximum standard deviation of the measured velocity for the slug flow was 0.010 m/s. Figure 20 presents the result of one of the repeated experiments, whose flow pattern was bubble flow, velocity was 0.313 m/s, and standard deviation for the 20 repeated experiments was 0.002 m/s. The results of the uncertainty evaluation show that the uncertainty of the developed system is small, and its repeatability is satisfactory.

## 5. Conclusions

In this work, a new contactless cross-correlation velocity measurement system was developed for the velocity measurement of gas–liquid two-phase flow in small channels. The contactless conductivity detection (CCD) sensor was introduced to obtain the conductance signal of the upstream and downstream fluid in a contactless way, and the cross-correlation velocity measurement principle was introduced to carry out the velocity measurement. To avoid the unfavorable influence of bubble/slug deformation and relative position changes, the developed new velocity measurement system has a compact three-electrode construction with one electrode reused. To ensure the synchronicity, consistency, and independence of the upstream sensor and downstream sensor, a switching unit that can achieve fast switching and time compensation was introduced.

To verify the effectiveness of the developed velocity measurement system, two typical experiments of the bubble flow and the slug flow were conducted using a prototype of the new contactless cross-correlation velocity measurement system with an inner diameter of 2.5 mm. The results of the two typical experiments show that the design of each part of the system was successful. The switching unit can effectively switch between the upstream CCD sensor and the downstream CCD sensor, and the conductance measurement units can effectively obtain the conductance of the upstream CCD sensor and the downstream CCD sensor. The upstream conductance signal and the downstream conductance signal have a significant correlation, and with the principle of cross-correlation velocity measurement, the velocity of the bubble/slug flow can be obtained.

To test the velocity measurement performance of the developed system, velocity measurement experiments were carried out for the bubble flow and slug flow using the prototype. The velocity range for the bubble flow was 0.312–0.816 m/s, and the maximum relative error of the flow rate measurement was 4.54%. The velocity range for the slug flow was 0.161–1.250 m/s, and the maximum relative error of the flow rate measurement was 3.70%. In addition, the uncertainty of the developed system was also evaluated. The standard deviation of 20 repeated velocity measurement experiments was obtained. The maximum standard deviation of the measured velocity for the bubble flow was 0.012 m/s, and the maximum standard deviation of the measured velocity for the slug flow was 0.010 m/s.

The flow measurement system developed in this paper combines the CCD technique with the principle of cross-correlation velocity measurement. Compared with the optical velocity measurement methods/systems, the developed system has better adaptability to nontransparent channels and fluids. Compared with the conventional contact electrical methods, the CCD electrodes are not in contact with the fluid directly, thereby avoiding electrochemical corrosion, electrode polarization, and flow interference. Unlike most of the existing CCD-based velocity measurement systems that require four electrodes, the velocity measurement system developed in this work uses a more compact three-electrode construction, and ensures the synchronicity, consistency, and independence by introducing a switching unit. In addition, although the three-electrode construction fulfils the solidified fluid assumption to the greatest extent possible, deformation and relative position changes still exist, and in this work, their influences on measurement were neglected. In the future, the influences of deformation and relative position changes on the conductance/impedance detection and velocity measurement will be investigated to further improve the measurement performance.

## Figures and Tables

**Figure 1 sensors-23-04886-f001:**
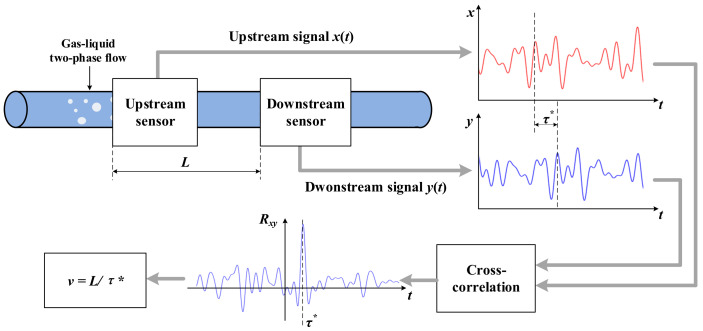
Measurement principle of the cross-correlation velocity.

**Figure 2 sensors-23-04886-f002:**
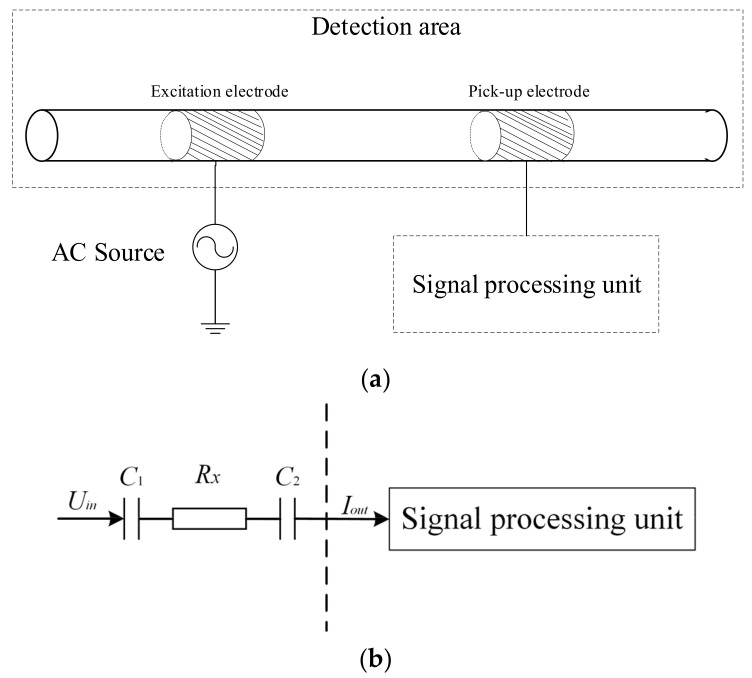
CCD sensor: (**a**) construction of the CCD sensor; (**b**) measurement principle of the CCD sensor.

**Figure 3 sensors-23-04886-f003:**
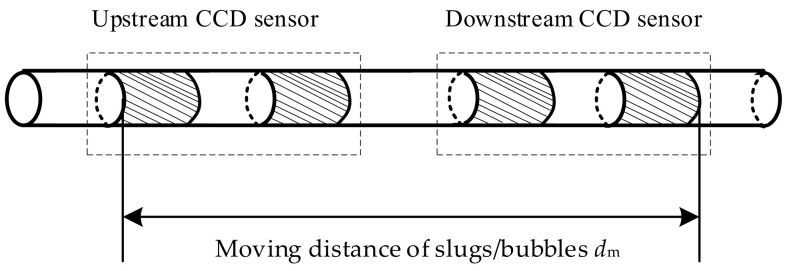
Moving distance of the slugs/bubbles $d_{m}$.

**Figure 4 sensors-23-04886-f004:**
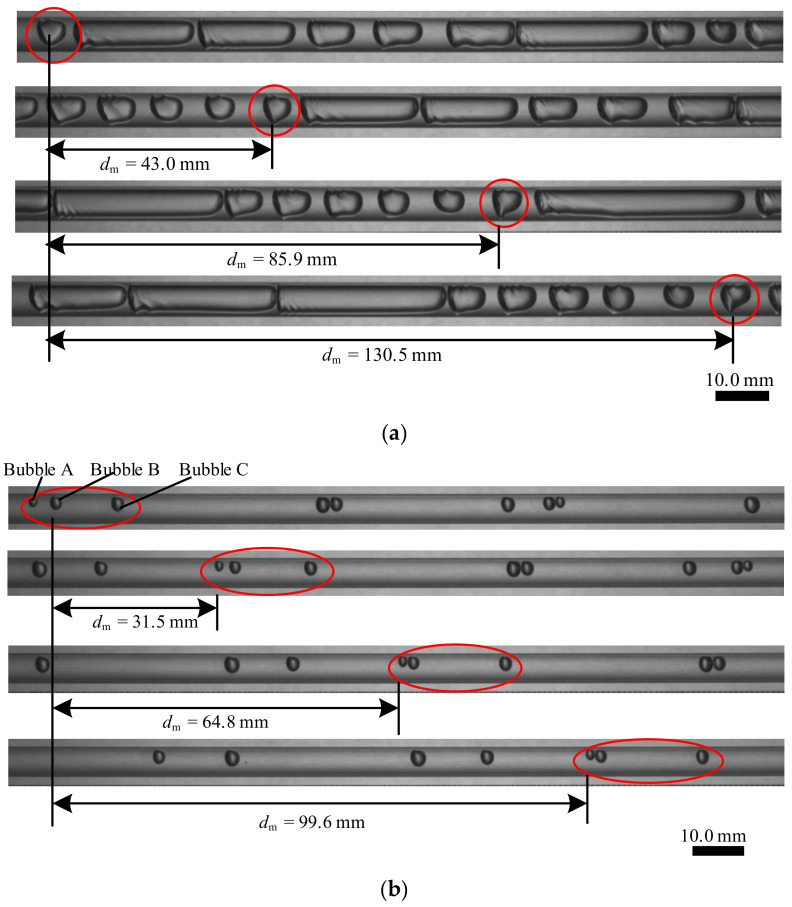
Typical example of deformation and relative position change: (**a**) deformation; (**b**) relative position change.

**Figure 5 sensors-23-04886-f005:**
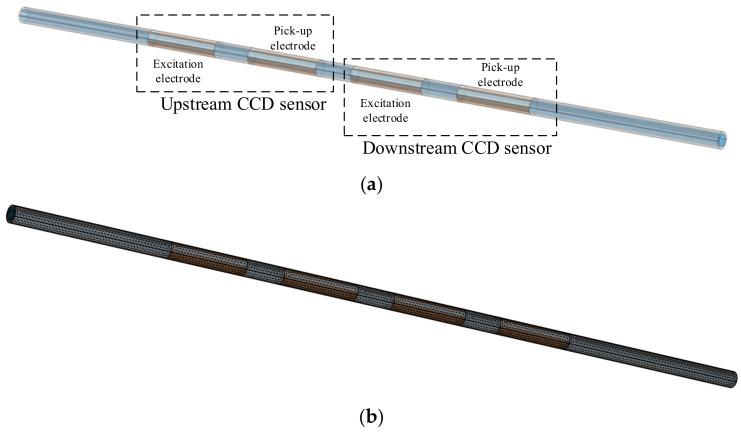
FEM model: (**a**) construction; (**b**) computational grid.

**Figure 6 sensors-23-04886-f006:**
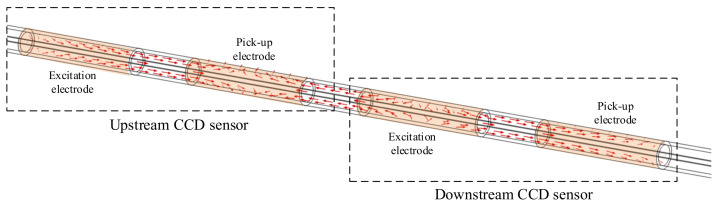
Simulation result of the two CCD sensors working together.

**Figure 7 sensors-23-04886-f007:**
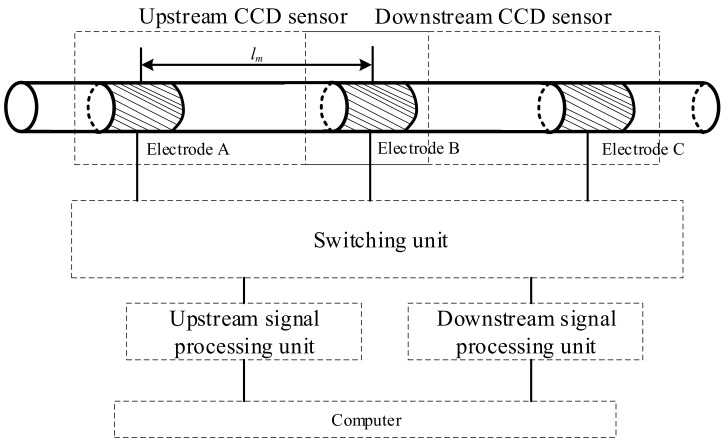
Construction of the velocity measurement system.

**Figure 8 sensors-23-04886-f008:**
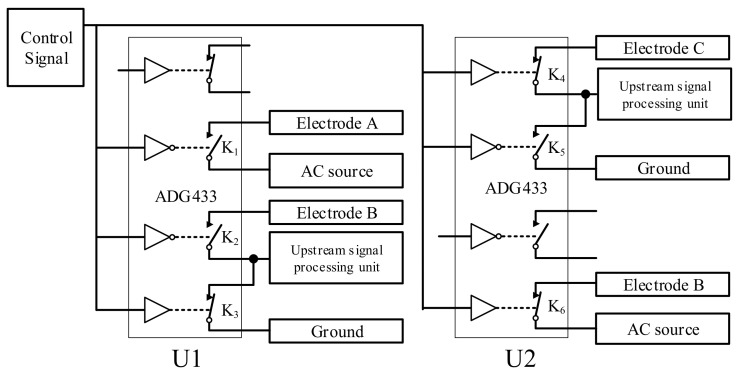
The circuit connections in the switching unit.

**Figure 9 sensors-23-04886-f009:**
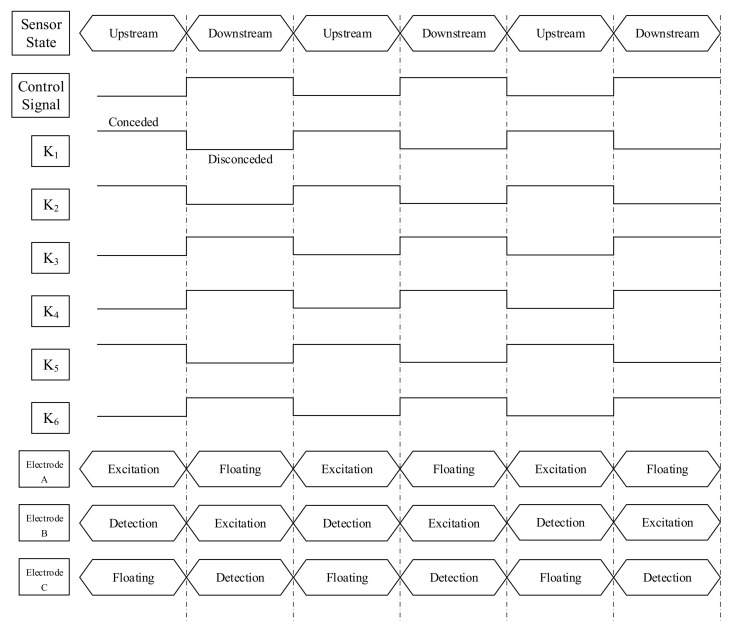
Sequence diagram of the switching unit.

**Figure 10 sensors-23-04886-f010:**
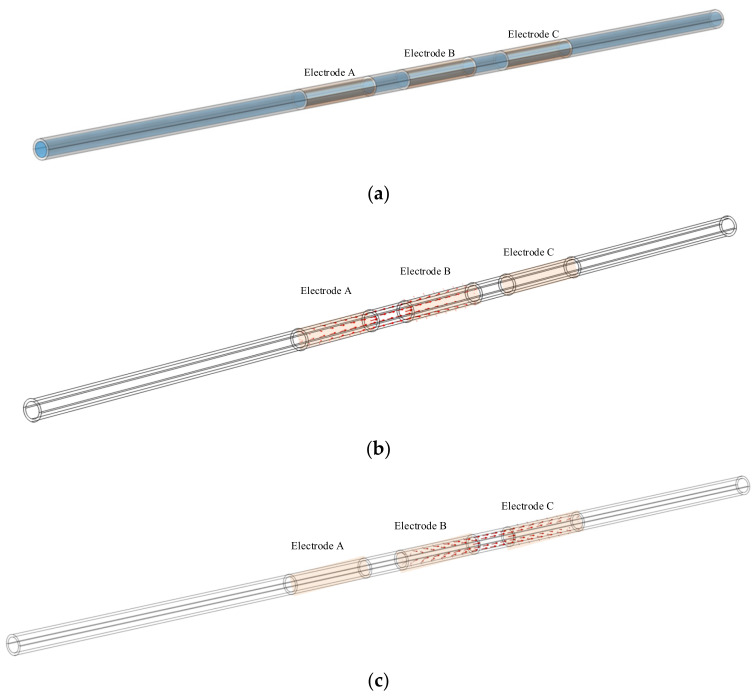
Simulation results for the three-electrode velocity measurement system: (**a**) simulation model; (**b**) current density simulation result for the upstream working mode; (**c**) current density simulation result for the downstream working mode.

**Figure 11 sensors-23-04886-f011:**
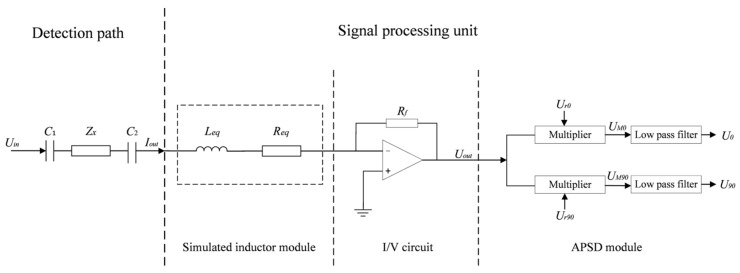
Conductance measurement unit.

**Figure 12 sensors-23-04886-f012:**
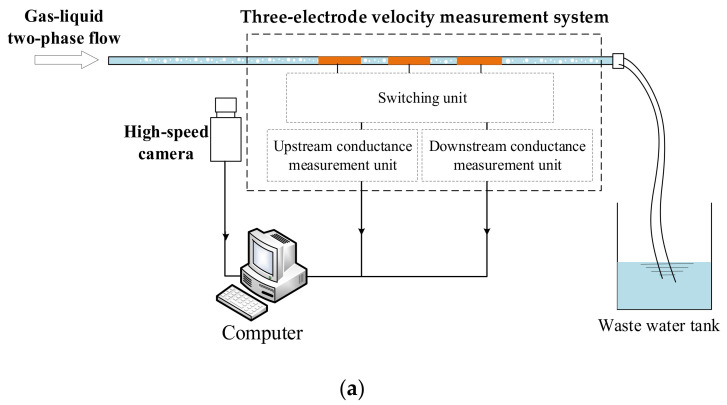
Experimental setup: (**a**) diagram of the experimental setup; (**b**) photo of the prototype.

**Figure 13 sensors-23-04886-f013:**
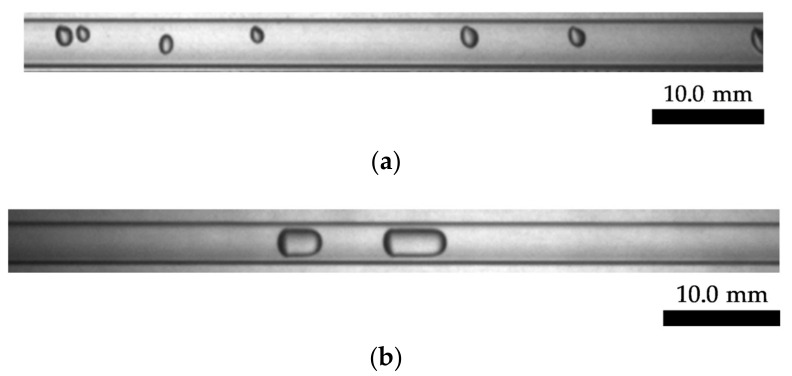
Photos of the experimental gas–liquid two-phase flow: (**a**) bubble flow; (**b**) slug flow.

**Figure 14 sensors-23-04886-f014:**
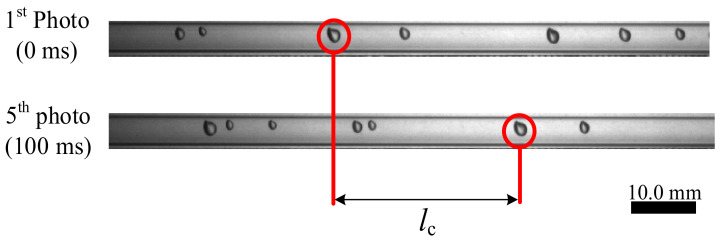
Example of obtaining the $v_{\mathrm{ref}}$.

**Figure 15 sensors-23-04886-f015:**
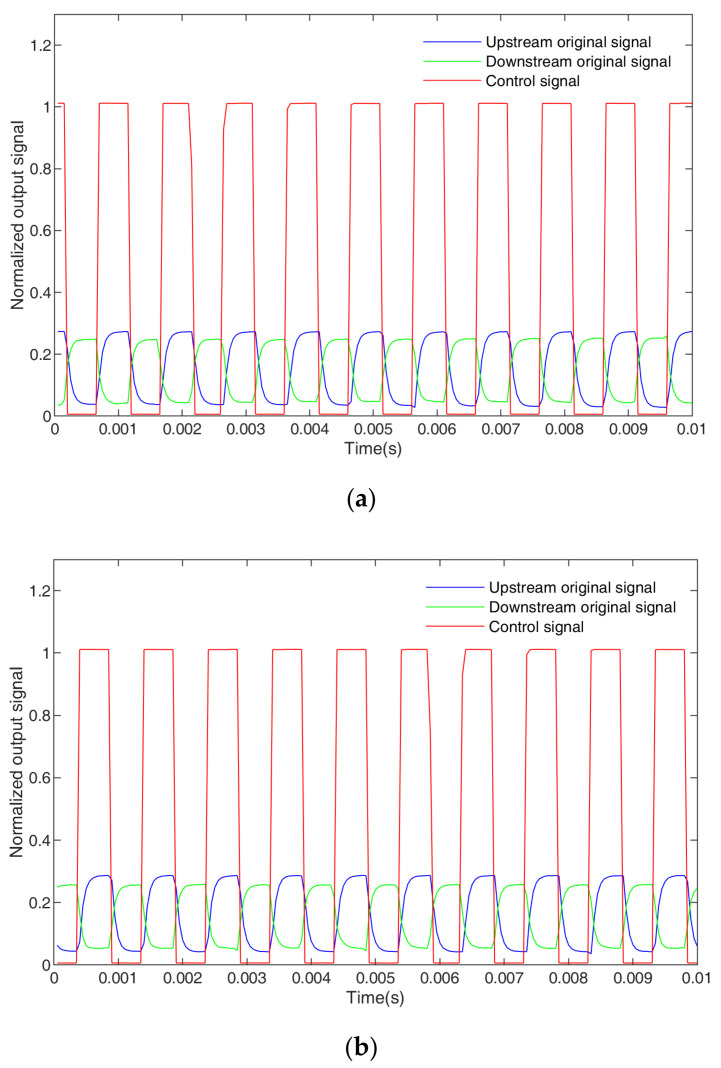
The output signal of the system: (**a**) bubble flow, (**b**) slug flow.

**Figure 16 sensors-23-04886-f016:**
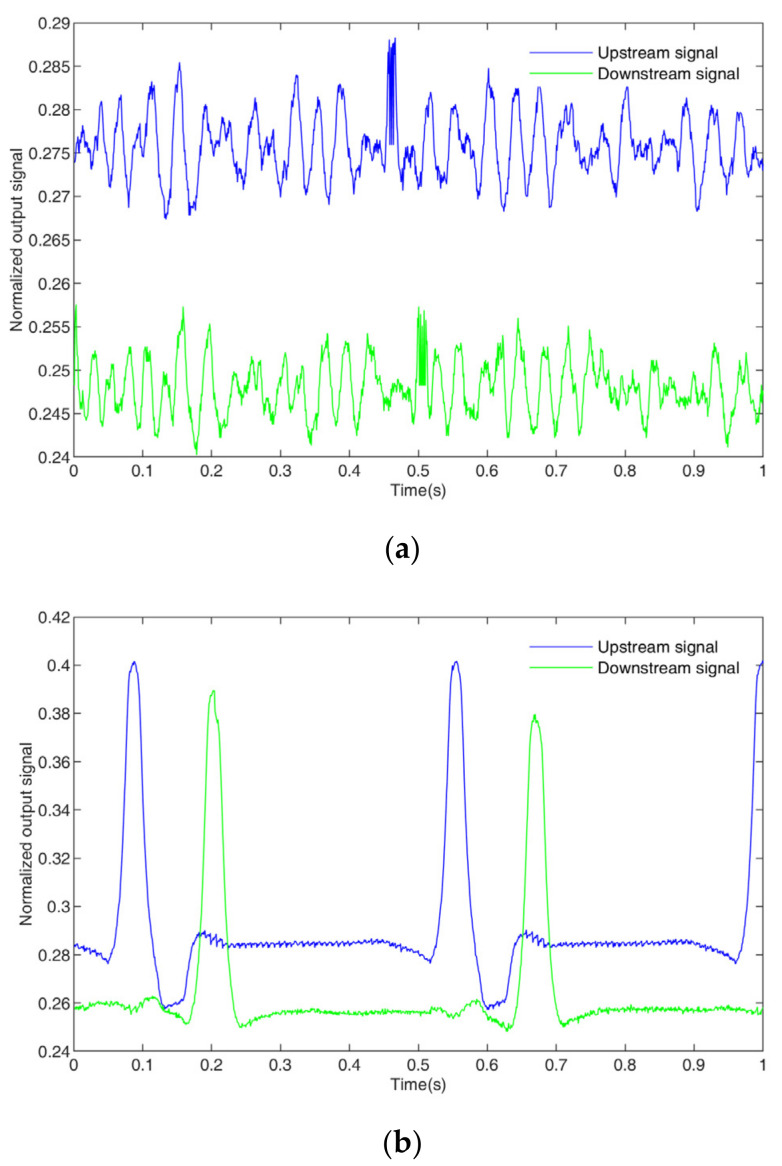
Original signal sequences: (**a**) bubble flow, (**b**) slug flow.

**Figure 17 sensors-23-04886-f017:**
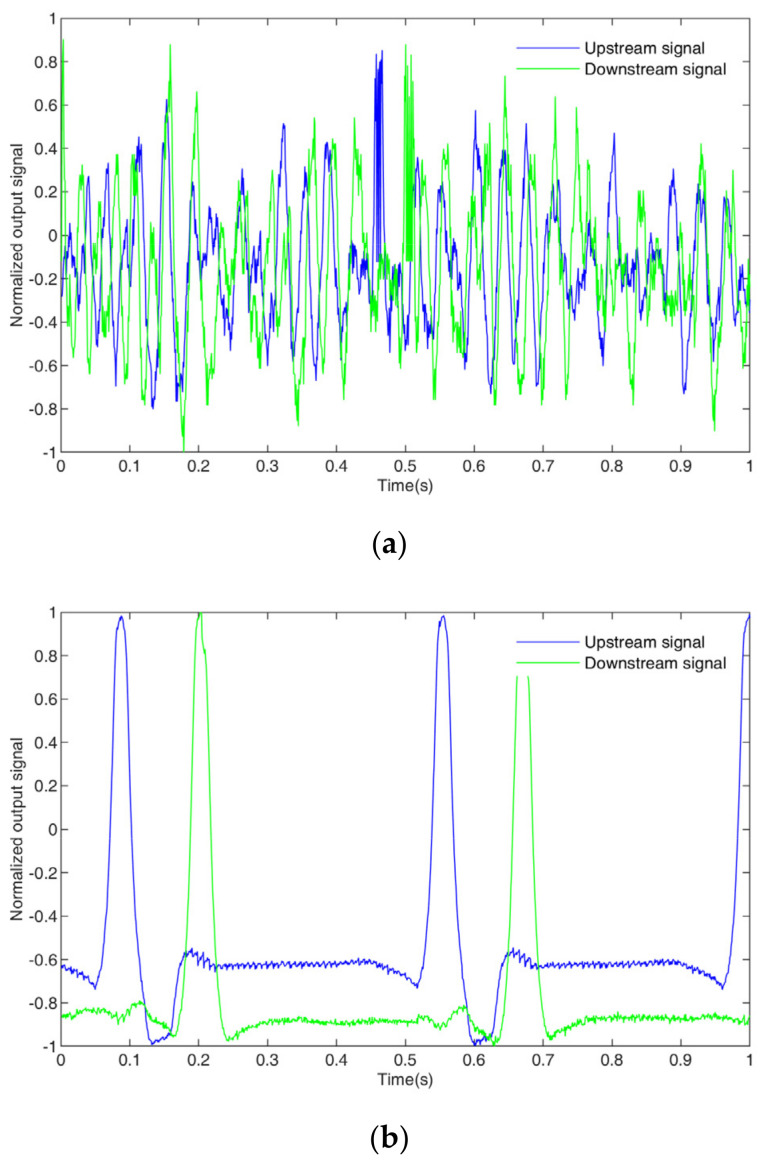
Standardized signal sequences: (**a**) bubble flow, (**b**) slug flow.

**Figure 18 sensors-23-04886-f018:**
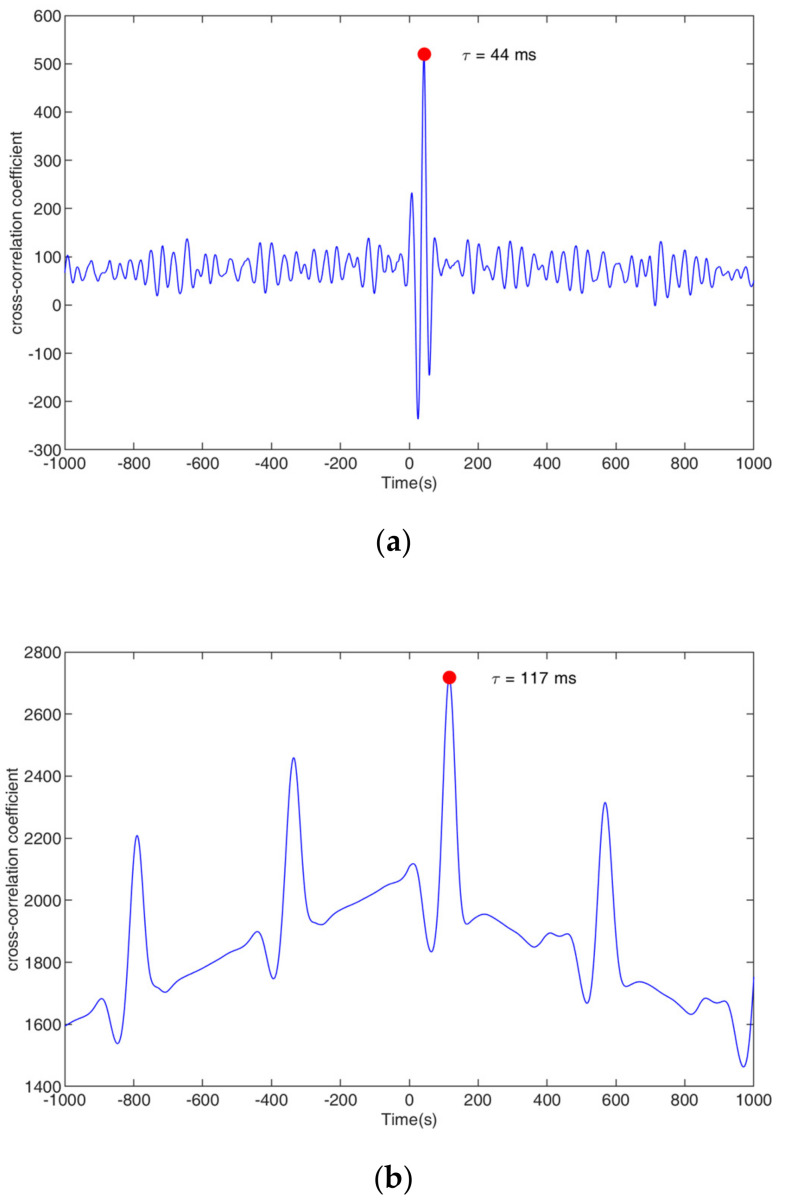
Cross-correlation coefficient and transit time: (**a**) bubble flow, (**b**) slug flow.

**Figure 19 sensors-23-04886-f019:**
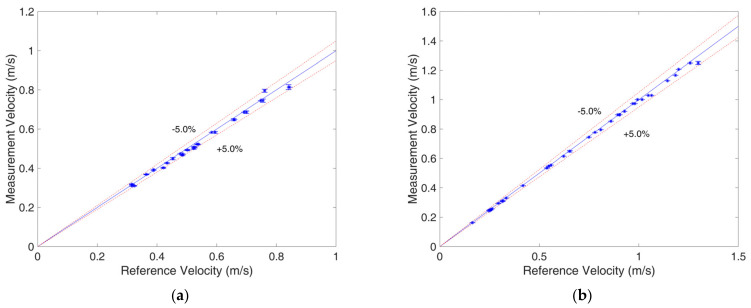
Velocity measurement result: (**a**) bubble flow, (**b**) slug flow.

**Figure 20 sensors-23-04886-f020:**
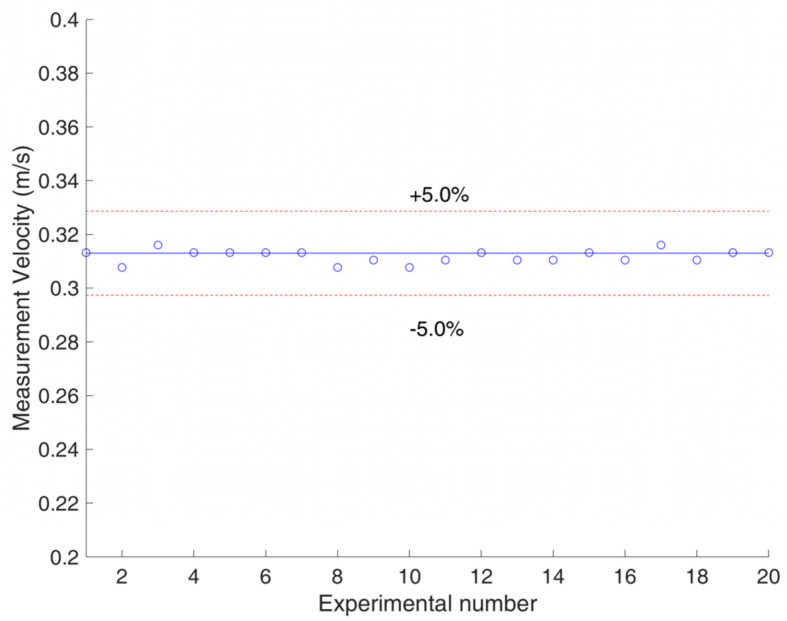
Result for one of the repeated experiments (flow pattern: bubble; velocity: 0.313 m/s).

**Table 1 sensors-23-04886-t001:** Relative positions of the bubbles when they are flowing.

$d_{m}$	Distance between A and B	Distance between B and C
0.0 mm	2.5 mm	9.4 mm
31.5 mm	1.3 mm	12.1 mm
64.8 mm	0.3 mm	15.5 mm
99.6 mm	0.2 mm	17.3 mm

**Table 2 sensors-23-04886-t002:** Properties of the model components.

Components	Materials	Conductivity (S/m)	Relative Permittivity
pipe wall	quartz glass	0	4.2
electrodes	copper	$5\times 10^{7}$	
external air	air	0	1
fluid	Tap water	0.01	78

**Table 3 sensors-23-04886-t003:** Parameters of the prototype.

Inner Diameter (mm)	2.50	Gap between B and C (mm)	10.03
Outer diameter (mm)	4.38	Distance between upstream and downstream (mm)	30.01
Length of electrode A (mm)	20.05		
Length of electrode B (mm)	19.96	Electrode material	Copper
Length of electrode C (mm)	20.01	Channel material	Glass
Gap between A and B (mm)	10.10	Fluid	Tap water/nitrogen

## Data Availability

The data presented in this study are available on request from the corresponding author. The data are not publicly available due to policy reasons.

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
