# Peer review of "A New Contactless Cross-Correlation Velocity Measurement System for Gas–Liquid Two-Phase Flow"

_sensors, 2023, doi:10.3390/s23104886_

Round 1

Reviewer 1 Report

The current study proposes an experimental method to measure the velocity of gas-liquid two-phase flow in channels without physical contact, utilizing a three-electrode velocity measurement system founded on the principle of contactless conductivity detection. The topic explored in this work is expected to capture the interest of the readership of the Sensors journal. Nevertheless, the paper displays certain shortcomings that need to be addressed prior to its suitability for publication in the Sensors journal.

- The originality of the current study is not evident in the manuscript. It would be beneficial to provide a more comprehensive explanation of the novel contributions of the research to clarify its significance to the scientific community.

- The introduction section of the manuscript appears to be deficient in terms of a literature review. To enhance the clarity and comprehensibility of the present work, the authors should conduct a thorough literature review that elucidates how the current research is rooted in prior studies. It is recommended that the authors discuss the relevance and importance of the present study in relation to previous research related to their work. Additionally, the following relevant studies should be included in the paper for discussion.

* Wang, C., Huang, J., Ji, H., & Huang, Z. (2022). Response Characteristics of Contactless Impedance Detection (CID) Sensor on Slug Flow in Small Channels: The Investigation on Slug Separation Distance. Sensors, 22(22), 8987.

* Bordbar, A., Kheirandish, S., Taassob, A., Kamali, R., & Ebrahimi, A. (2020). High-viscosity liquid mixing in a slug-flow micromixer: a numerical study. Journal of Flow Chemistry, 10, 449-459.

* Tang, X. Y., Huang, J., Ji, H., Wang, B., & Huang, Z. (2020). New method for bubble/slug velocity measurement in small channels. Review of Scientific Instruments, 91(5), 055001.

- The authors are encouraged to provide a detailed explanation of the distinctions between the current work and their prior publications on this topic (as noted in the two references provided). Furthermore, the authors should clarify the original contributions of the current study in comparison to the previous works and their own publications. It is crucial to explicitly delineate the novel aspects of the present research and how they differ from previous findings. 

*Guo, Z., Huang, J., Huang, Q., Jiang, Y., Ji, H., & Huang, Z. (2020). New Contactless Velocity Measurement Sensor for Bubble/Slug Flow in Small Scale Pipes. IEEE Access, 8, 198035-198046. 

* Huang, J., Ji, H., Huang, Z., Wang, B., & Li, H. (2017). A new contactless method for velocity measurement of bubble and slug in millimeter-scale pipelines. IEEE Access, 5, 12168-12175.

- The authors should state the assumptions made to develop the finite element model.

- The authors should provide a detailed description of the finite element model employed in the study. This includes the numerical schemes and procedures utilized, the boundary conditions set, the fluid properties used in the simulations, and the computational grid size and configuration. These details are crucial for readers to comprehend the methodology and procedures employed in the study, and to evaluate the accuracy and reliability of the results. 

- The authors should consider analyzing, reporting, and discussing the uncertainties associated with the experimental measurements in the paper. Evaluating the uncertainties is essential to determine the reliability and accuracy of the experimental results, and to validate the experimental setup and instrumentation. It is recommended that the authors conduct a thorough uncertainty analysis of the experimental measurements, including a description of the sources of uncertainty and the methods used to estimate the uncertainties.

- Error bars should be added to the figures presented in the paper.

- A scale bar should be added to Figures 13 and 14.

- Figure 12(b) looks blurred. The authors should improve the quality of the figure.

- The manuscript in its present form lacks a sufficient discussion on the results. The authors should consider discussing all the results in detail, highlighting the key findings and explaining their implications. In addition, the authors should explore the limitations of the study and provide suggestions for future research.

- Conclusions are a summary of the manuscript. The conclusion section of the manuscript should provide a concise summary of the key findings and their significance in the context of the research question. It is recommended that the authors revise this section to provide a clear and comprehensive overview of the research outcomes.

- There are inconsistencies in the use of English. It is recommended that a native English speaker reviews and corrects the paper.

- There are inconsistencies in the use of English. It is recommended that a native English speaker reviews and corrects the paper.

Author Response

The current study proposes an experimental method to measure the velocity of gas-liquid two-phase flow in channels without physical contact, utilizing a three-electrode velocity measurement system founded on the principle of contactless conductivity detection. The topic explored in this work is expected to capture the interest of the readership of the Sensors journal. Nevertheless, the paper displays certain shortcomings that need to be addressed prior to its suitability for publication in the Sensors journal.

Comment 1 - The originality of the current study is not evident in the manuscript. It would be beneficial to provide a more comprehensive explanation of the novel contributions of the research to clarify its significance to the scientific community.

Response 1: Thanks for the reviewer’s suggestion. The flow measurement system developed in this paper combines CCD technique with the principle of cross-correlation velocity measurement. Compared with the optical velocity measurement methods/systems, the developed system has higher adaptability to non-transparent channels and fluids. Compared with the conventional contact electrical methods, the electrodes of CCD are not in contact with the fluid directly, thereby avoiding the electrochemical corrosion, electrode polarization, and flow interference. Unlike most existing CCD-based velocity measurement systems that require four electrodes, the velocity measurement system developed in this work uses a more compact three-electrode construction, and ensures the synchronicity, consistence and independence by introducing a switching unit. As suggested, in the revised manuscript the corresponding statement has been added to clarify the novel contributions of this work, which could be found in Line 620-629, Page 21.

Comment 2- The introduction section of the manuscript appears to be deficient in terms of a literature review. To enhance the clarity and comprehensibility of the present work, the authors should conduct a thorough literature review that elucidates how the current research is rooted in prior studies. It is recommended that the authors discuss the relevance and importance of the present study in relation to previous research related to their work. Additionally, the following relevant studies should be included in the paper for discussion.

* Wang, C., Huang, J., Ji, H., & Huang, Z. (2022). Response Characteristics of Contactless Impedance Detection (CID) Sensor on Slug Flow in Small Channels: The Investigation on Slug Separation Distance. Sensors, 22(22), 8987.

* Bordbar, A., Kheirandish, S., Taassob, A., Kamali, R., & Ebrahimi, A. (2020). High-viscosity liquid mixing in a slug-flow micromixer: a numerical study. Journal of Flow Chemistry, 10, 449-459.

* Tang, X. Y., Huang, J., Ji, H., Wang, B., & Huang, Z. (2020). New method for bubble/slug velocity measurement in small channels. Review of Scientific Instruments, 91(5), 055001.

Response 2: As suggested, in the revised manuscript new references have been added (reference [12], [21] and [43]) and the discussions that concern on relationship between the previous research and this work has also been added, which could be found in Line 36-41, Page 1, Line 63-73, Page 2 and Line 112-116, Page 3. Thanks for the reviewer’s suggestion which provides us useful references and help us enhance the clarity and comprehensibility of the present work.

Comment 3- The authors are encouraged to provide a detailed explanation of the distinctions between the current work and their prior publications on this topic (as noted in the two references provided). Furthermore, the authors should clarify the original contributions of the current study in comparison to the previous works and their own publications. It is crucial to explicitly delineate the novel aspects of the present research and how they differ from previous findings. 

*Guo, Z., Huang, J., Huang, Q., Jiang, Y., Ji, H., & Huang, Z. (2020). New Contactless Velocity Measurement Sensor for Bubble/Slug Flow in Small Scale Pipes. IEEE Access, 8, 198035-198046. 

* Huang, J., Ji, H., Huang, Z., Wang, B., & Li, H. (2017). A new contactless method for velocity measurement of bubble and slug in millimeter-scale pipelines. IEEE Access, 5, 12168-12175.

Response 3: As suggested, in the revised manuscript, a detailed explanation of the distinctions between the current work and our prior publications on the velocity measurement by CCD has been added, which could be found in Line 105-116, Page 3.

Comment 4 - The authors should state the assumptions made to develop the finite element model.

Response 4: As suggested, in the revised manuscript the assumptions made to develop the finite element model has been added, which could be found in Line 200-202, Page 6.

Comment 5 - The authors should provide a detailed description of the finite element model employed in the study. This includes the numerical schemes and procedures utilized, the boundary conditions set, the fluid properties used in the simulations, and the computational grid size and configuration. These details are crucial for readers to comprehend the methodology and procedures employed in the study, and to evaluate the accuracy and reliability of the results. 

Response 5: As suggested, in the revised manuscript the more detailed description of the finite element model employed has been added, which could be found in Line 188-215, Page 6 and Line 330-340, Page 10. And a new figure which illustrates the computational grid is also added, which could be found in Figure 5(b), Page 6.

Comment 6- The authors should consider analyzing, reporting, and discussing the uncertainties associated with the experimental measurements in the paper. Evaluating the uncertainties is essential to determine the reliability and accuracy of the experimental results, and to validate the experimental setup and instrumentation. It is recommended that the authors conduct a thorough uncertainty analysis of the experimental measurements, including a description of the sources of uncertainty and the methods used to estimate the uncertainties.

Response 6: As suggested, in the revised manuscript, the experimental part has been expanded, new experiments are carried out to evaluate the uncertainties, and the corresponding statements about analyzing, reporting, and discussing could be found in Line 572-588, Page 19-20.

Comment 7 - Error bars should be added to the figures presented in the paper.

Response 7: As suggested, in the revised manuscript the error bars have been added in the corresponding figures, which could be found in Figure 19, Page 19.

Comment 8 - A scale bar should be added to Figures 13 and 14.

Response 8: As suggested, in the revised manuscript the scale bars have been added in the corresponding figures, which could be found in Page 14.

Comment 9 - Figure 12(b) looks blurred. The authors should improve the quality of the figure.

Response 9: As suggested, in the revised manuscript the Figure 12(b) has been replaced by a higher quality figure, which could be found in Figure 12 (b), Page 13.

Comment 10 - The manuscript in its present form lacks a sufficient discussion on the results. The authors should consider discussing all the results in detail, highlighting the key findings and explaining their implications. In addition, the authors should explore the limitations of the study and provide suggestions for future research.

Response 10: Thanks for the reviewer’s suggestion, in the revised manuscript, more discussions on the experimental results have been added in the experimental section, which could be found in Line 545-588, Page 19-20. And the limitations of the study and suggestions for future research have been added in the Conclusion Section, which could be found in Line 629-634, Page 21.

Comment 11 - Conclusions are a summary of the manuscript. The conclusion section of the manuscript should provide a concise summary of the key findings and their significance in the context of the research question. It is recommended that the authors revise this section to provide a clear and comprehensive overview of the research outcomes.

Response 11: Thanks for your suggestion, in the revised manuscript, the Section ‘Conclusion’ has been reorganized according to the reviewer’s suggestions to make the research outcomes/results clearer, which could be found in Line 590-634, Page 20-21.

Comment 12 - There are inconsistencies in the use of English. It is recommended that a native English speaker reviews and corrects the paper.

Response 12: As suggested, the authors have checked the consistencies and the grammar of the manuscript and invited a British colleague to check the manuscript.

Reviewer 2 Report

With interest I have read the authors work describing their new micro fluidic measurement device utilising a three electrodes sensor in series and the principle of cross correlation. 

Firstly, the authors even though at parts they touch on some of these aspects, they should be far more clear about:

  1. Any existing gap in the literature (as established with newer research - eg with more recent references from 2022 and 2023)
  2. How their work aims to address this gap (saying that a new design will be implemented is not enough - also just mentioning the principles of the design is desirable, but also not enough on its own: the authors need to expand on the above points, but also discuss the way this search of the best sensor satisfying these principles can be carried out in a manner that is convincing to the readers). 
  3. Elaborate on how the (3) principles (synchronicity, consistence and independence) their sensor aims to satisfy are distinct or similar to what is done in the past from other researchers. This will be a good opportunity for the authors to expand on the technical meaning and definition of those principles (which will be useful to be done when these principles are first mentioned. 

There are many grammar errors repeatedly appearing in the text and many at important parts of the text too. 

There are many case but the ones that repeatedly stand out the most are the cause of “synchronous, consistent and independent ” instead of “synchronicity, consistence and independence”.

For example (not exhaustive) see grammar mistakes in lines 73, 116. 121, 140, 165, 

Also the use of some words is awkward eg use “describe” instead of “descript”. Also avoid using “And” at the start of a sentence. 

Section 2: should the text be restructured to focus on shaping a narrative for presenting this as the challenges that need be considered to specify an efficient sensor design?

Figure 4 also shows the case of bubbles merging. How are these cases treated by the sensor developed and how do such cases affect the errors observed? If such coalescence of bubbles happens to form longer ones would this affect the optimal distance between the electrodes? How are such considerations that embrace the range of variable conditions in practical cases, could be taken into account to provide confidence for the use of the sensor or likewise appreciate any possible limitations for any such extreme cases?

It’s interesting to see the use of computational methods in sections 2.2 and first sections of 3. for demonstration purposes but it’s not clear if there has been a search for identifying the optimal spacing distance between the electrodes and results around why the chosen design is the best performing one. 

Further in section 3.1: Focus on elaborating further how this design choice (is it a first random choice based on the principles or the one that best satisfies them?) may help most efficiently satisfy the required sensor design principles mentioned before. 

The above results should then be validated experimentally for a range of flow conditions   (Eg different flow velocities) in addition to the two types of flow conditions tried. 

Specifically section 4.3 is more of a validation section than just a results demonstration (so the title needs to also be more focused and read as “4.3 Experimental validation” than “results”) and would be nice to see this validation more expanded for different flow conditions that just those tried. 

Last, errors for this sensor are mentioned. Are these expected to be changing with the flow conditions in addition to the type of flow?

More discussion is needed on this to offer confidence for the utility of this device in a range of multiphase flows (specifically expand on ranges of velocities and pipe diameters/flow geometry) and specify as clearly as possible the limitations of the device (eg that the results hold for flow conditions similar to those tried therein). 

Please see my comments above for grammar. 

Author Response

With interest I have read the authors work describing their new micro fluidic measurement device utilising a three electrodes sensor in series and the principle of cross correlation. 

Comment 1: Firstly, the authors even though at parts they touch on some of these aspects, they should be far more clear about:

  1. Any existing gap in the literature (as established with newer research - eg with more recent references from 2022 and 2023)
  2. How their work aims to address this gap (saying that a new design will be implemented is not enough - also just mentioning the principles of the design is desirable, but also not enough on its own: the authors need to expand on the above points, but also discuss the way this search of the best sensor satisfying these principles can be carried out in a manner that is convincing to the readers). 
  3. Elaborate on how the (3) principles (synchronicity, consistence and independence) their sensor aims to satisfy are distinct or similar to what is done in the past from other researchers. This will be a good opportunity for the authors to expand on the technical meaning and definition of those principles (which will be useful to be done when these principles are first mentioned. 

Response 1: As suggested, in the manuscript more recent references have been added, including the Reference [17], [18], [19], [20] and [43].

Meanwhile, according to the research work of this work, the gap of the existing works is summarized to show the significance of this work, which could be found in Line 63-73, Page 2, Line 105-116, Page 3 and Line 612-620, Page 20-21 of the revised manuscript.

The mentioned three principles are necessary for velocity measurement based on the principle of cross-correlation velocity measurement, whatever the type of the sensors used. In the conventional works, the velocity measurement system/method is mainly developed for normal scale channels, and the sensors are optical sensor, acoustic/ultrasonic sensors and contact electrical sensors. These sensors have strong directivity, so the synchronous, consistent and independent measurement can be easily realized. With the channel scale decreasing to the small channel, the adaptability of the above measurement methods also decreases, while CCD sensors are suitable for measurement in small channels. However, when CCD is used for velocity measurement, it will face challenges in satisfying the synchronicity, consistence and independence. In the revised manuscript, the corresponding description could be found in Line 140-148, Page 4.

Comment 2: There are many grammar errors repeatedly appearing in the text and many at important parts of the text too. 

There are many case but the ones that repeatedly stand out the most are the cause of “synchronous, consistent and independent ” instead of “synchronicity, consistence and independence”.

For example (not exhaustive) see grammar mistakes in lines 73, 116. 121, 140, 165, 

Also the use of some words is awkward eg use “describe” instead of “descript”. Also avoid using “And” at the start of a sentence. 

Response 2: Thanks for the reviewer’s time and help, the mentioned grammar errors have been modified and the authors carefully checked the whole text for grammatical errors.

Comment 3: Section 2: should the text be restructured to focus on shaping a narrative for presenting this as the challenges that need be considered to specify an efficient sensor design?

Response 3: As suggested, in the revised manuscript, the discussion in the Section 2.1 and Section 2.2 are modified and summarized, and a new subsection ‘Section 2.3’ has been added to describe the challenges of designing an Effective Velocity Measurement System based on CCD, which could be found in Line 149-153, Page 4, Line 182-188, Page 6 and Line 242-256, Page 7-8. Thanks for the reviewer’s suggestion which improve the readability of the paper.

Comment 4: Figure 4 also shows the case of bubbles merging. How are these cases treated by the sensor developed and how do such cases affect the errors observed? If such coalescence of bubbles happens to form longer ones would this affect the optimal distance between the electrodes? How are such considerations that embrace the range of variable conditions in practical cases, could be taken into account to provide confidence for the use of the sensor or likewise appreciate any possible limitations for any such extreme cases?

Response 4: As the reviewer mentioned, bubbles merging is existed, in this work, we regard the bubbles merging as a case of deformation. In the velocity measurement, the deformation (including the bubbles merging) will lead to impedance change and further affect the cross-correlation result. The longer the distance of bubble/slug movement, the greater the possibility of the deformation occurrence. Moreover, for gas-liquid two-phase flow with deformation, an increased distance of bubble movement results in greater degree of deformation, which leads to larger changes in equivalent conductivity. Therefore, this paper implements a three-electrode design to minimize the occurrence of such case. If obvious deformation does occur within the measurement range (total 80mm), it can lead to the failure of flow rate measurement. Determining the extreme flow rate at which deformation is too obvious to cause cross-correlation failure involves considering several other factors, such as the conductivity of the liquid phase, the volume fraction of phases, and the flow viscosity. We apologize for not being able to provide the judges with such information in a short time. It will require further research to investigate, thanks for your understanding. Meanwhile, in the revised manuscript, the theoretical upper and low limit of the measurement system in the absence of deformation has been added, which could be found in Line 528-542, Page 18.

Comment 5: It’s interesting to see the use of computational methods in sections 2.2 and first sections of 3. for demonstration purposes but it’s not clear if there has been a search for identifying the optimal spacing distance between the electrodes and results around why the chosen design is the best performing one. 

Response 5: More detailed information about the FEM model has been added in the revised manuscript, including the assumptions made to develop the model, the boundary conditions set, the fluid properties, and the computational grid configuration, which could be found in Line 188-215, Page 6 and Line 330-340, Page 10.

Comment 6: Further in section 3.1: Focus on elaborating further how this design choice (is it a first random choice based on the principles or the one that best satisfies them?) may help most efficiently satisfy the required sensor design principles mentioned before. 

Response 6: This is not a "first random choice". Its design principle is as follows: firstly, at least two identical sensors are required based on the principle of cross-correlation, and each CCD sensor must have at least two electrodes based on the CCD measurement principle. Therefore, by reusing electrodes, the implementation of two CCD sensors using three electrodes is currently the solution that requires the fewest electrodes and the least space. It should be noted that this is currently the best solution found, but it cannot be guaranteed as the best choice. In the revised manuscript, the design principle of the present three-electrode constructions of the velocity measurement system has been added, which could be found in Line 267-273, Page 8.

Comment 7: The above results should then be validated experimentally for a range of flow conditions   (Eg different flow velocities) in addition to the two types of flow conditions tried. 

Specifically section 4.3 is more of a validation section than just a results demonstration (so the title needs to also be more focused and read as “4.3 Experimental validation” than “results”) and would be nice to see this validation more expanded for different flow conditions that just those tried. 

Response 7: Thanks for the reviewer’s suggestion, as suggested, in the revised manuscript, new experiments with different flow conditions (different flow velocities) are carried out. Section 4.3 has been expanded with the experimental results, which could be found in Line 545-562, Page 19-20.

Comment 8: Last, errors for this sensor are mentioned. Are these expected to be changing with the flow conditions in addition to the type of flow?

More discussion is needed on this to offer confidence for the utility of this device in a range of multiphase flows (specifically expand on ranges of velocities and pipe diameters/flow geometry) and specify as clearly as possible the limitations of the device (eg that the results hold for flow conditions similar to those tried therein). 

Response 8: Thanks for the reviewer’s suggestion, as suggested, in the revised manuscript, the measurement performance of the system (measurement errors) and the measurement uncertainty have been discussed, which could be found in Line 563-588, Page 19-20.

It is necessary to indicate that, this paper focuses on the designing of a velocity measurement system, and the experiments only focuses on velocity measurement within a certain range for two flow patterns. We can guarantee the measurement performance of the designed flow measurement system within the experimental flow rate range. Further research on the influence of different flow conditions on the measurement will be our future works. Thank you for your understanding.

Round 2

Reviewer 1 Report

The authors have addressed the reviewers' comments in the revised version of the manuscript. The manuscript can be considered for publication after editorial corrections.

There are spelling and grammatical errors in the text that should be corrected.

Reviewer 2 Report

I want to thank the authors for putting the effort to address most of my feedback.

English language can always benefit the manuscript presentation, by proofreading the manuscript again.